# The Genetic Diversity and Genetic Structure of the Germplasm Resources of the Medicinal Orchid Plant *Habenaria dentata*

**DOI:** 10.3390/genes14091749

**Published:** 2023-09-01

**Authors:** Yishan Yang, Jianmin Tang, Rong Zou, Yajin Luo, Zhenhai Deng, Dongxin Li, Shengfeng Chai, Xiao Wei

**Affiliations:** 1Guangxi Key Laboratory of Plant Functional Substances and Resources Sustainable Utilization, Guangxi Institute of Botany, Guilin 541006, China; yangyishan0113@163.com (Y.Y.); ldx@gxib.cn (D.L.); csf@gxib.cn (S.C.); wx@gxib.cn (X.W.); 2The United Graduate School of Agricultural Sciences, Kagoshima University, Kagoshima 890-0065, Japan; 3Guangxi Yachang Orchid National Nature Reserve Management Center, Baise 533200, China; luoyajin320@163.com (Y.L.); 68368598@163.com (Z.D.)

**Keywords:** *Habenaria dentata*, medicinal orchid plant, genetic diversity, ISSR, cluster analysis

## Abstract

*Habenaria dentata* has medicinal and ornamental value, but the number of wild populations is decreasing dramatically. Thus, conducting research on its genetic diversity and structure is necessary to provide a basis for its conservation. This study aimed to explore the genetic diversity of the wild plant *H. dentata* and protect and optimize its wild resources. The genetic diversity of 133 samples from six wild populations of *H. dentata* was analyzed using Inter Simple Sequence Repeat molecular markers to provide a scientific basis for the screening of improved germplasm resources. The results showed that the average number of alleles was 1.765, the average number of effective alleles was 1.424, the average Nei’s gene diversity index was 0.252, the average Shannon diversity index was 0.381, and the average percentage of polymorphic loci was 76.499%. The variation within the populations was 77.34%, and the variation between the populations was 22.66%. The gene flow was 1.705, which was greater than 1. The results of the cluster analysis showed that the six populations were mainly divided into four clusters and were not classified according to their geographical location. There was no significant correlation between the geographical location and genetic distance between the populations (r = 0.557, *p* > 0.05). The genetic diversity of *H. dentata* is high. Among the six wild populations, the genetic diversity of the Mulun population was the highest and this population can be used as a key protection unit. The study on the genetic diversity of *H. dentata* can not only reveal the reasons for the decrease in the number of individuals in the population to a certain extent, and put forward the protection strategy, but also provide a scientific basis for the breeding of excellent seed resources.

## 1. Introduction

As one of the largest families of angiosperms, Orchidaceae is a highly evolved group with a wide distribution in all the terrestrial ecosystems of the world except for the polar regions and deserts [1]. However, because most orchids require specific mycorrhizal fungi for survival and seed germination [2,3] and more specific growth environments, they have a narrower distribution and are vulnerable to external environmental influences [4]. In addition, many orchids also have high ornamental [5] and medicinal values [6], and their wild plant resources are often removed, reducing the variety and number of species. In the Guangxi Zhuang Autonomous Region, there are 510 species of orchids in the wild, accounting for 31.88% of the total number of orchid species in China [7]. *H. dentata*, which has high medicinal and ornamental value, is a terrestrial herb of the genus *Habenaria*, family Orchidaceae, which mainly grows in gullies or hillside woodlands at an altitude of 190–2300 m. The plant is widely distributed in China, such as Guangdong, Guangxi, Anhui, Fujian, Yunnan, Guizhou, and other areas in southwest China [8]. Additionally, this plant is also a medicinal plant, with its oval or oblong fleshy underground tubers being used as medicine. This medicinal material is mild in nature, sweet in taste, slightly bitter, and has degassing, detoxication [9], diuretic, and anti-inflammatory [10] effects and relieves coughs and phlegm [11]. It can also be used for the treatment of body deficiency, orchitis, impotence, tuberculosis cough, hernia, urinary tract infections, and other diseases. Although *H. dentata* is widely distributed in China, the number of individuals in a single population in the field is small and the distribution is scattered. In addition, because of its medicinal and ornamental value, the number of wild populations is decreasing dramatically. Based on the above reasons, it is necessary to conduct research on this medicinal plant resource with great development potential to ensure its protection.

Currently, the research on *H. dentata* has mainly focused on resource investigation [12], tissue culture technology [13,14], and mycorrhizal fungi [15,16,17], but its conservation genetics and genetic breeding have not been studied. The determination of genetic diversity is one of the first goals for plant germplasm resource protection, and it is also a solid basis for managing the utilization and development of medicinal plant resources. Genetic diversity research mainly explores the diversity differences of the DNA, proteins, chromosomes, and phenotypes [18]. The genetic variation between different populations of the same species and between different individuals in the same population is often the main focus of genetic diversity research [19]. A study on the genetic diversity of medicinal plant germplasm resources can not only provide information on the genetic structure and gene polymorphism of the populations but can also suggest potential directions for the selection of the best germplasm resources and core protected populations.

Molecular markers are one of the basic methods for the analysis of species’ genetic diversity. Molecular marker techniques, including inter-simple sequence repeats (ISSRs) and simple sequence repeats (SSRs), can identify the genetic differences between species and are not affected by environmental factors [20]. ISSRs have the advantages of simple operation, being low cost, and having high polymorphism. Therefore, the ISSRs are widely used in genetic diversity research [21,22,23] and the phylogenetic analysis [24] of orchids. To the best of our knowledge, this is the first study to focus on the use of ISSRs to explore the reasons for the decline in the population size and genetic diversity of *H. dentata.* Therefore, in order to reveal the reasons for the decline in the number of individuals and the genetic diversity of the population, this paper used ISSRs to analyze the genetic diversity and genetic relationship among 133 samples of six wild populations of the medicinal plant *H. dentata.* The findings can provide an important theoretical basis for the screening of excellent germplasm resources and the determination of key protection units at the molecular level.

## 2. Materials and Methods

### 2.1. Test Materials

Mature, clean, and fresh *H. dentata* leaves that were free of pest and disease contamination were collected from 1 to 10 September 2021 from each field population according to the population size and number of individuals at the collection site. The collected fresh leaves were placed in molecular sample bags and quickly dried with varicolored silica gel. The samples were sealed in bags and brought back to the laboratory, where they were stored in a cool and dry place. The details of the sample collection sites and the number of individuals that were collected from each field population are shown in Table 1. The six populations were distributed in areas with sufficient light and deep soil layers, among which only the Minqiang population (MQ) was distributed in the hilltop grasses. The Jianfeng Mountain population (JFS), the Zhu’eshang village population (ZES), and the Longji village population (LJ) were all located in the mountains or on the slopes of roadsides, where there was sufficient light and the terrain was conducive to drainage. When compared with the other populations, the Leye population (LY) and the Mulun population (ML) had more fertile soils, and they were located on hillside roadsides. Additionally, the ML population was located on river valley grasses. The population photos are provided in Figure 1.

### 2.2. Methodology

#### 2.2.1. Extraction and Quality Assay of the Total DNA

Washed forceps and disposable small steel beads were placed in a washed mortar, and an appropriate amount of anhydrous ethanol was poured into it and ignited for sterilization. After the sterilized equipment was at room temperature, about 500 mg of the dried leaves of *H. dentata* were placed in a sterile 2 mL centrifuge tube with the forceps, and then the appropriate amount of Polyvinylpyrrolidone (PVP) and 3–4 small steel balls were added. Then, the centrifuge tube was placed in a bead mill to crush the sample. Next, the total DNA was extracted from *H. dentata* using a modified CTAB method [18]. The concentration and quality of the total DNA were detected using a 1.2% agarose gel (with 8 µL of 4S Green Plus non-toxic nucleic acid dye) electrophoresis and micro-UV spectrophotometry, and finally, the extracted DNA was stored at −20 °C.

#### 2.2.2. Synthesis and Screening of the Inter-Simple Sequence Repeat Primers

The ISSR primers were synthesized based on the 100 universal primers that were published by Columbia College, Canada, and they were synthesized by Bioengineering Co., Ltd. (Shanghai, China). The genomic DNA of 30 samples (six wild populations, with five samples randomly selected from each population) was used as a template, and 100 ISSR primers were used for the polymerase chain reaction (PCR) pre-amplification. The primers with clear bands and good polymorphism were screened out for the subsequent genetic diversity analysis of the six wild populations of *H. dentata* [25].

#### 2.2.3. Inter-Simple Sequence Repeat Polymerase Chain Reaction System and Procedure

The PCR conditions were based on Lu et al. [26] and were as follows: a 25 µL total reaction volume containing 1 µL of the 10 µM primers, 1 µL of the DNA template, 0.6 µL of Taq, 2.5 µL of 10 × Taq Buffer, and 1.5 µL of 25 mM Mg^2+^, and the remaining volume was made up with ddH_2_O. The PCR reaction procedure was as follows: pre-denaturation at 95 °C for 5 min, 40 cycles of 95 °C for 20 s, 52 °C for 20 s, and 72 °C for 2 min, which was followed by an extension at 72 °C for 20 min to obtain the PCR amplification products. The amplification products were placed on a 1.2% agarose gel (with 8 µL of 4S Green Plus non-toxic nucleic acid dye), electrophoresed for 40 min, and photographed in a UV gel imaging system.

#### 2.2.4. Data Processing

Using the electrophoresis spectrum of the DL 2000 marker as a reference, the bands were manually reviewed according to the electrophoretic mobility of each band. The clear bands were recorded as “1”, while those with missing or weak bands were recorded as “0”, and the original binary data matrix of the 0/1 format was established using Excel 2019 software [27]. The “0/1 matrix” was imported into the POPGene 32 software to calculate the genetic diversity index of six populations, including the number of polymorphic loci, number of alleles (*N_a_*), effective number of alleles (*N*_e_), Nei’s genetic diversity (*H*_e_), Shannon diversity index (*I*), percentage of polymorphic loci (*PPL*), total genetic diversity (*H*_t_), genetic diversity among populations (*H*_s_), gene differentiation coefficient (*G*_st_), and gene flow (*N*_m_) [28]. The analysis of molecular variance (AMOVA) function in the WinArl 35 software was used to analyze the degrees of freedom (df), total variance, variation component, and total variation percentage. The genetic distance (GD) and genetic similarity (GS) were calculated using the NTSYS 2.10 e software, and a genetic similarity coefficient heat map was drawn using the Orange 3 software [28]. The “0/1 matrix” was imported into the NTsys 2.10e software, and the unweighted pair-group method with arithmetic means (UPGMA) in the software was used to establish a cluster tree [29]. Then, GenAlex 6.502 software was used for a principal component analysis (PCA) of all the individuals [30]. The geographical distance between the populations was calculated using the latitude and longitude of each collection site. The correlation between the genetic distance and geographical location among the populations was analyzed using the Mantel Test in the TFPGA software 1.3 [31].

## 3. Results

### 3.1. Detection of the Total DNA Quality

Five samples were randomly selected from the six populations, and the quality of the DNA was detected. The results showed that the extracted DNA bands were clear and there was no obvious tailing phenomenon. The quality of the DNA basically ranged from 80 to 400 ng, and the OD260/280 ranged from 1.7 to 2.0. The results showed that the total DNA had few impurities, was of good quality, and could be used for subsequent genetic diversity tests.

### 3.2. Primer Screening

The samples of *H. dentata* were screened with several rounds of primers, and eight primers with high amplification success were selected. The details of the primers are shown in Table 2. Figure 2 shows the amplification results of some of the samples that were carried out with primers 808 and 844.

### 3.3. Inter-Simple Sequence Repeat Genetic Diversity Analysis

#### 3.3.1. Analysis of the Genetic Diversity Parameters

The number of fragments that were obtained for each of the eight primers ranged between four and eight, the size of the fragments was basically 250–1000 bp, and the statistical results of the variation in the six populations are shown in Table 3. The table shows that at the species level, the *PPL* in *H. dentata* was 100%, the *I* was 0.493, the *H*_e_ was 0.324, the *N*_e_ was 1.538, and the *N_a_* was 2.00. At the population level, six populations had no significant differences in the *N*_a_, *N*_e_, *H*_e_, *I*, or *PPL* (*p* > 0.05), and the genes of the six wild populations were evenly distributed across the group.

The *PPL* of the six wild populations ranged from 69.23 to 84.62%, with a mean value of 76.497%, and the highest number of polymorphic loci was found in the ML population and the lowest was found in the JFS and MQ populations. The *N_a_* ranged from 1.692 to 1.846, with a mean value of 1.765. The highest number of alleles was found in the ML population, while the lowest number of alleles was found in the JFS and MQ populations. The *N_e_* ranged from 1.353 to 1.506, with a mean value of 1.424, of which the ML population had the highest *N_e_* and the JFS population had the lowest. The *I* ranged from 0.321 to 0.436 with a mean value of 0.381, with the highest value for the ML population and the lowest for the JFS population. The highest genetic diversity was found in the ML population and the lowest was found in the JFS population.

#### 3.3.2. Population Genetic Structure Analysis

The *H_e_* of the populations of *H. dentata* is shown in Table 4. The *H*_t_ was 0.326 ± 0.019, the *H*_s_ was 0.252 ± 0.010, and the inter-population genetic diversity (*H*_t_ − *H*_s_) was 0.074. Moreover, in the six wild populations of *H. dentata*, the intra-population variation (*H*_s_/*H*_t_) was 77.3% and the inter-population variation (*H*_t_ − *H*_s_/*H*_t_) was 22.7%, indicating that only 22.7% of the variation was among the populations and 77.3% was within the populations. The *N*_m_ was 1.706 (>1), indicating that the gene flow between the populations was more frequent and less prone to genetic differentiation. Furthermore, the results of the AMOVA analysis are presented in Table 5. It is evident from the table that 27% of the genetic variation occurred between populations and 73% occurred within populations. The *H_e_* and the AMOVA analysis were in general agreement, indicating that the genetic variation in *H. dentata* mainly occurred within the populations and that there was little genetic differentiation between the populations.

#### 3.3.3. Genetic Distance and Clustering Analysis of *H. dentata*

As shown in Figure 3 and Figure 4, the genetic distances of the JFS and LY populations were the smallest, and the genetic distances of the ZES and MQ populations were the largest. The genetic concordance of the ZES and MQ populations was the smallest, and the genetic concordance among the ZES and LY populations was the largest.

#### 3.3.4. Cluster Analysis of the Wild Populations

As shown in Figure 5, the six *H. dentata* populations were mainly divided into four branches, with the ML and JFS populations clustering together, the LY and ZES populations clustering together, and these two populations clustering with the MQ and LJ populations. Further testing using the Mantel Test showed that there was no significant correlation between the geographical location and the genetic distance between the populations (r = 0.557, *p* > 0.05).

#### 3.3.5. Principal Component Analysis of the Populations

The genetic distances among the 133 individuals of the six populations were analyzed using a PCA, and PCA plots were constructed for individuals in the six *H. dentata* populations (Figure 6). As can be seen from the figure, the difference represented by the horizontal axis (principal component [PC]-1) was 13.31% and the difference represented by the vertical axis (PC-2) was 24.54%. The LJ population was mainly distributed above the horizontal axis, and most of the individuals in this population were distant from the individuals in the other populations. The MQ population was mostly distributed to the left of the *y*-axis, and most of the individuals were distant from the other clusters. The majority of the individuals in the LY and ZES populations were closer together and converged on the right side of the *y*-axis. The majority of the individuals of the ML and JFS populations were distributed below the *x*-axis, and the individuals of the two clusters were closer to each other. The results of the principal component analysis were similar to those of the UPGMA clustering map.

## 4. Discussion

In general, the higher the genetic diversity of a population, the better its germplasm resources [32], and the less the individuals in the population are affected by other external factors, the more adapted they are to the environment and the faster the expansion of the population. In contrast, populations or species with a lower genetic diversity are less adaptable and more likely to be endangered [33]. Populations with a high genetic diversity can be used for the selection and breeding of individuals to promote the genetic diversity of the species, which is especially important for the conservation of endangered species. The level of genetic diversity of a species is often evaluated based on the three parameters, namely the *PPL*, *H_e_*, and *I*. In this study, the ISSR molecular marker technique was used for the first time to analyze the genetic diversity of six wild populations of *H. dentata* in the Guangxi Zhuang Autonomous Region, and the results showed that the *PPL* was 100%, *I* was 0.493, and *H_e_* was 0.324. For instance, Huang et al. [24] used ISSR molecular marker technology to study the genetic diversity of the germplasm resources of the jewel orchid, *Anoectochilus roxburghii*, and the results of the study indicated that the *PPL* was 96.51%, *I* was 0.424, and *He* was 0.272. Then, Li et al. [34] examined *Paphiopedilum micranthum*, and the *PPL* of the seven populations was 96.04%, the *I* was 0.488, and the *H_e_* was 0.324. Additionally, Jiang et al. [35] selected 12 populations of *Cymbidium kanran* in the wild and found that the genetic diversity was high at the species level, and the *PPL* was 78.90%, *I* was 0.399, and *H_e_* was 0.264. Through comparison, it was found that the genetic diversity of *H. dentata* was higher than that of *A. roxburghii*, *P. micranthum,* and *C. kanran*. The highest genetic diversity at the population level was found in the ML population, and it is recommended that this population is used as a key conservation unit and for the selection and breeding of the species.

The genetic differentiation of populations can be divided into three levels according to the size of the *N*_m_, with *N*_m_ > 1 being a high amount of gene flow, 0.25–0.99 being a medium amount, and 0–0.249 being a low amount of gene flow [36,37]. In the case of high levels of gene flow, genetic differentiation due to genetic drift is small, and with medium and low levels of gene flow, due to less gene exchange between populations, genetic drift is more likely to occur and leads to an increased genetic differentiation. Referring to this criterion, the *N*_m_ of *H. dentata* was 1.706 (>1), which is a high level, supporting that *H. dentata* populations have more frequent gene exchange and are less likely to have genetic differentiation among the populations. Furthermore, the *G*_st_ was 0.227, indicating that only 22.7% of the variation was among the populations and 77.3% was within the populations. The AMOVA analysis also showed that only 27% of the genetic variation occurred among the populations and 73% of the genetic variation occurred within the populations. It has been shown that the genetic differentiation coefficient is 0.59 for self-incompatible taxa, 0.23 for heterozygous taxa, and 0.19 for mixed taxa, and, therefore, *H. dentata* is probably a heterozygous or mixed taxon plant [38].

The high level of gene flow among populations is closely related to the long-distance dispersal of seeds. As orchids produce numerous tiny seeds, once they are released into the air, they can be spread over a long distance with the wind [39,40]. However, the actual propagation distance of the seeds is highly dependent on the surrounding vegetation. The denser the vegetation, the shorter the seed propagation distance, and the more limited the gene exchange between populations [41]. For instance, Lee et al. [42] studied *Habenaria linearifolia* and found that it is mainly located on a plateau that is surrounded by dense tall grasses. This growth environment limits seed dispersal and thus hinders gene exchange. Whereas, *H. dentata* is mainly distributed in hillsides or woodlands. Consequently, the seeds can be transmitted by the wind over a long distance, which increases the gene exchange between populations and is conducive to the accumulation of the population genetic diversity.

In this study, the genetic diversity of the ML and LY populations was the highest. Both populations were in nature reserves and were far away from human development. Because of fewer human activities, these populations have been better protected. However, the genetic diversity of the MQ population, which is also in protected areas and far from human development, was low. Therefore, it is speculated that there may be two reasons for this finding. (1) The MQ population is distributed in the mountain hilltop grass, and the dense tall grass hinders gene exchange between the populations. (2) The establishment time of this population was recent, and therefore, the genetic diversity may have not had time to accumulate. It has been suggested that populations with a short establishment time have usually not yet accumulated high genetic diversity through gene mutations or gene flow with other populations [43,44]. The analysis of the genetic diversity of *H. dentata* also has a high reference value for the study of its population evolution history. It is mainly distributed in subtropical and warm temperate zones, and it is also rarely distributed in the Korean Peninsula. Researchers have found that the allozymes of the population of *H. dentata* on the Korean Peninsula are not diverse and the genetic diversity of the population is low. Thus, the distribution of *H. dentata* on the Korean Peninsula may have originated from a single ancestral population from southern China or southern Japan, which migrated to the Korean Peninsula from one or more refuges after the last glacial maximum [45].

Currently, it is increasingly believed that genetic distances between different populations of the same species do not necessarily lead to genetic variation and that factors such as genetic mutations, selection, and gene flow are the main causes of genetic differentiation between populations [46]. For example, Qin et al. [47] analyzed the genetic diversity of a very small population and eight wild populations of *P. emersonii* based on genetic distance, and the results showed that the six populations were not clustered strictly according to their geographical location and they were clustered according to individual differences. There was also no significant correlation between the genetic distance of the geographical location and the clusters (r = 0.557, *p* > 0.05). Therefore, this paper suggests that even though there is frequent genetic exchange between populations in close proximity, strong genetic mutations and selection can lead to genetic differentiation between populations.

The results of this study showed that the gene flow among the populations of *H. dentata* in Guangxi, China, was frequent, the genetic differentiation among the populations was high, and abundant genetic diversity was accumulated in the long-term evolutionary process. Therefore, it is suggested that the reason for the decline in the number of individuals in the population may not be due to a lack of genetic diversity, but rather due to extensive harvesting or the destruction of their natural habitats. Based on the results of this study, the following suggestions are proposed to strengthen the protection of the wild resources of *H. dentata*. (1) The seeds of different populations of *H. dentata* in different regions (provinces) should be collected as much as possible to establish a germplasm resource bank with mycorrhizal fungi to protect the genetic diversity. (2) For small-scale wild populations, it is recommended that measures should be taken to protect them in situ, and we suggest the establishment of natural protection zones to prevent the occurrence of poaching. (3) Then, individuals with high genetic diversity should be used for breeding improved varieties, realizing artificial sowing and planting, and increasing the number of populations. Although the findings of this study have provided a foundation for SSR or random amplified polymorphic DNA analysis methods, this study only used ISSRs to analyze the genetic diversity of this population, and the PCR amplification system has not been optimized. Furthermore, few polymorphic primers were screened out for genetic diversity analysis. To obtain more comprehensive genetic information, future studies should expand the sampling area, increase the sampling volume, and use a variety of molecular markers. This study combined the analysis and evaluation of the genetic diversity and genetic structure of the wild resources of *H. dentata* in China, and the findings provide a theoretical basis for the protection of the plant resources of *H. dentata*.

## Figures and Tables

**Figure 1 genes-14-01749-f001:**
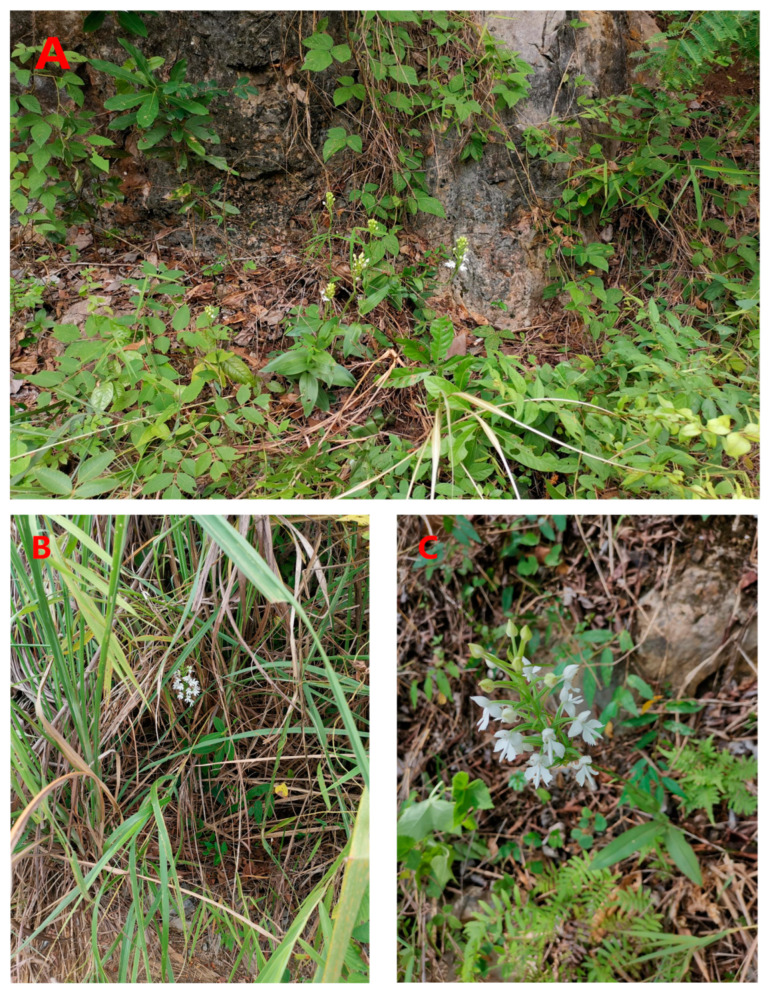
Partial population photos of *H. dentata. (***A**) Jianfeng Mountain (JFS) population (Mountain slope), (**B**) Minqiang (MQ) population (Hilltop grass), and (**C**) Leye (LY) population (Roadside grass).

**Figure 2 genes-14-01749-f002:**
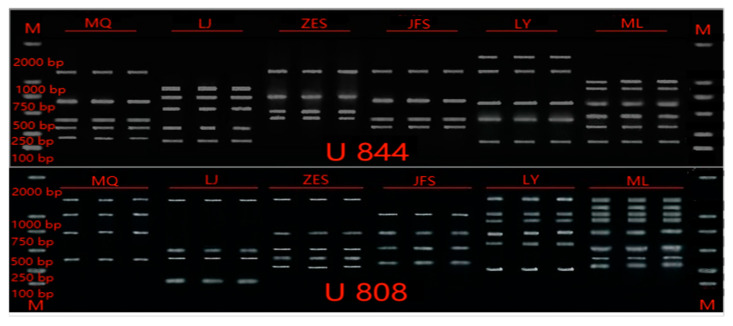
Amplification of primers U808 and U844 for 18 samples from eight populations.

**Figure 3 genes-14-01749-f003:**
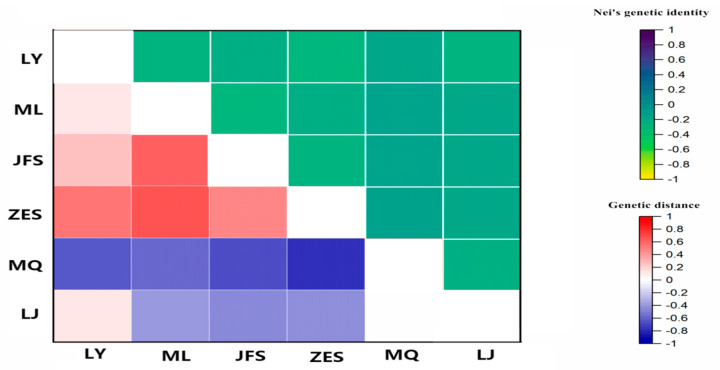
The heat map of the genetic distance (below the diagonal) and Nei’s genetic identity (above the diagonal) of the six populations. Abbreviations: MQ, the Minqiang population; JFS, the Jianfeng Mountain population; ZES, the Zhu’eshang village population; LJ, the Longji village population; LY, the Leye population; and ML, the Mulun population.

**Figure 4 genes-14-01749-f004:**
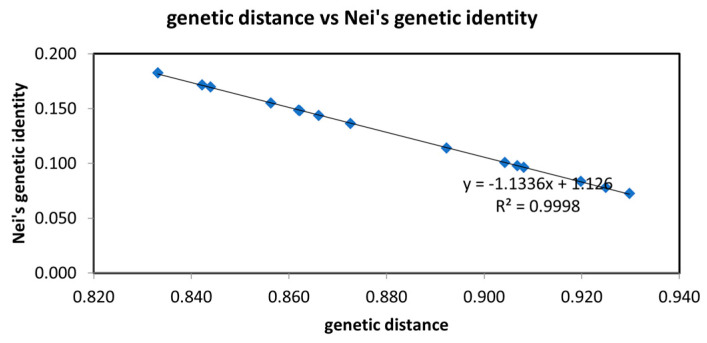
Linear analysis of the genetic distance and genetic similarity coefficient of the six populations.

**Figure 5 genes-14-01749-f005:**
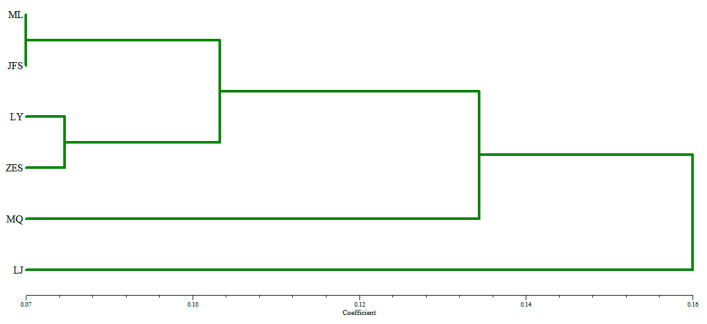
Dendrogram of the unweighted pair-group method with arithmetic means cluster analysis based on Nei’s genetic distance between the six populations of *H. dentata*. Abbreviations: MQ, the Minqiang population; JFS, the Jianfeng Mountain population; ZES, the Zhu’eshang village population; LJ, the Longji village population; LY, the Leye population; and ML, the Mulun population.

**Figure 6 genes-14-01749-f006:**
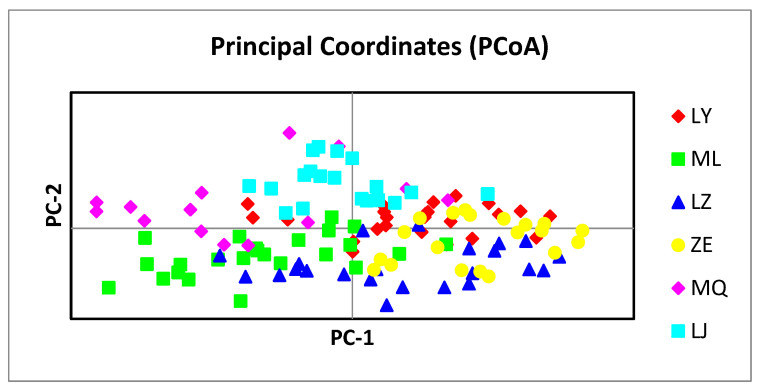
Principal coordinate analysis of the individuals of six populations of *H. dentata.* Abbreviations: MQ, the Minqiang population; JFS, the Jianfeng Mountain population; ZES, the Zhu’eshang village population; LJ, the Longji village population; LY, the Leye population; and ML, the Mulun population.

**Table 1 genes-14-01749-t001:** Information on the collection sites of *H. dentata*.

Area	North Latitude	East Longitude	Number of Samples	Biotope
Minqiang Village, Shanglong Township, Longzhou County, Chongzuo City (MQ)	22°25′23″	106°54′43″	22	Hilltop grass
Jianfeng Mountain, Xinzhuangxiong Village, Jiangzhou District, Chongzuo City (JFS)	22°29′30″	106°55′2″	23	Mountain slopes
Jingxi City, Lu Dong Township (ZES)	23°8′00″	106°19′45″	20	Roadside slopes
Longji Village, Luodong Township, Jingxi City (LJ)	23°8′22	106°21′26″	20	Roadside slopes
Huaping Village, Huaping Township, Leye County, Baise City (LY)	24°50′55″	106°21′50″	24	Roadside grasses
Huanjiang Maonan Autonomous County, Hechi City, Chuanshan Town, Xizaitun Mu Lun Nature Reserve (ML)	24°55′33″	106°33′42″	24	Grass by the river

**Table 2 genes-14-01749-t002:** Amplification results of the eight inter-simple sequence repeat primers.

Primer	Primer Sequences (5′-3′)	Annealing Temperature (°C)	Number of Total Amplified Bands	Number of Polymorphic Bands	Polymorphic Percentage (%)
U808	AGAGAGAGAGAGAGAGGC	48.7	8	8	100%
U823	TCTCTCTCTCTCTCTCTCC	48.0	5	4	80.0%
U844	CTCTCTCTCTCTCTCTCTRC	48.6	6	6	100%
U855	ACACACACACACACACACACYT	52.7	7	5	71.4%
U876	GATAGATAGACAGACA	38.4	6	4	66.7%
U878	GGATGGATGGATGGAT	47.0	7	6	85.7%
U880	GGAGAGGAGAGGAGA	47.9	6	5	83.3%
U885	BHBGAGAGAGAGAGAGAGA	48.0	5	4	80.0%

**Table 3 genes-14-01749-t003:** Genetic variation in the different populations of *H. dentata*.

Population	Number Of Polymorphic Loci	Number of Alleles (*N*_a_)	Effective Number of Alleles (*N*_e_)	Nei’s Genetic Diversity (*H*_e_)	Shannon’s Information Index (*I*)	Percentage of Polymorphic Loci (*PPL*)
ML	33	1.846 ± 0.366	1.506 ± 0.362	0.294 ± 0.184	0.436 ± 0.256	84.621
LY	31	1.795 ± 0.409	1.463 ± 0.350	0.275 ± 0.1807	0.414 ± 0.251	79.492
JFS	27	1.692 ± 0.468	1.353 ± 0.361	0.211 ± 0.195	0.321 ± 0.275	69.232
ZES	30	1.769 ± 0.427	1.387 ± 0.356	0.235 ± 0.1806	0.361 ± 0.252	76.922
LJ	31	1.795 ± 0.409	1.430 ± 0.353	0.257 ± 0.1828	0.393 ± 0.250	79.493
MQ	27	1.692 ± 0.468	1.406 ± 0.365	0.240 ± 0.1949	0.360 ± 0.277	69.234
Mean	29.8	1.765	1.424	0.252	0.381	76.499
*p* value		0.168	0.101	0.087	0.089	
Species level		2.000	1.539	0.324	0.493	100

Abbreviations: MQ, the Minqiang population; JFS, the Jianfeng Mountain population; ZES, the Zhu’eshang village population; LJ, the Longji village population; LY, the Leye population; and ML, the Mulun population.

**Table 4 genes-14-01749-t004:** The genetic diversity in *H. dentata*.

	Total Genetic Diversity (*H*_t_)	Population Genetic Diversity (*H*_s_)	Gene Differentiation Factor (*G*_st_)	Gene Flow (*N*_m_)
Mean	0.326	0.252	0.227	1.706
Standard deviation	0.019	0.010		

**Table 5 genes-14-01749-t005:** Analysis of molecular variance for the populations of *H. dentata*.

Source of Variation	df	Sum of Squares	Variance Components	Variation Percentage	*p* Value
Among populations	5	250.856	2.014	27%	<0.01
Within populations	127	708.618	5.580	73%	<0.01
Total	132	959.474	7.594	100%	

Abbreviation: df, degrees of freedom.

## Data Availability

All the data that were cited in the study are publicly available.

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
