# Peer review of "The Genetic Diversity and Genetic Structure of the Germplasm Resources of the Medicinal Orchid Plant Habenaria dentata"

_genes, 2023, doi:10.3390/genes14091749_

Round 1

Reviewer 1 Report

This manuscript reports the genetic diversity and genetic relationship among 133 samples of 6 wild populations of medicinal plant Habenaria dentata by ISSR markers. The aims of the manuscript (to analyze the genetic diversity and genetic relationship) are not in accordance with the content, since the authors also analyzed population genetic structure. However, the most serious problem of the study is to have performed the analyses using ISSR markers. ISSR markers are discontinued dominant markers, almost obsolete, just like RAPD. In fact, I find it strange that the authors claim that ISSR markers "are widely used in genetic diversity, identification of germplasm resources and genetic relationship of orchids", having cited only two manuscripts for justification, and of fair quality. Although ISSR markers are inexpensive, it is not comparable to the analysis of genetic diversity and differentiation of populations by codominant markers such as AFLP, allozyme, microsatellite and RFLP, among others. Currently, co-dominant markers have also greatly lowered their costs. Moreover, the great advantage of codominant markers is their ability to distinguish between homozygotes and heterozygotes, which cannot be achieved by dominant markers.

The study of population genetic diversity from co-dominant markers in Habenaria dentata and other species such as H. radiata and H. lineariforia are very well documented in populations from South Korea and Japan, however, this information is not considered by the authors. For the same reason, I am surprised that the authors have omitted important work done by Chung et al 2018 (https://doi.org/10.1007/s12229-017-9190-5), Chung et al 2018 (https://doi.org/10.5091/plecevo.2018.1366) and Lee et al 2022 (https://doi.org/10.3389/fpls.2022.772621), which would be highly recommended to replicate their methodology in Chinese populations.

I suggest the authors to conduct their studies with codominant markers instead of dominant markers in order to obtain more robust and reliable results.

Author Response

Dear reviewer,

  In view of your valuable comments, we have made the following changes to the manuscript. The first, in this manuscript, the genetic structure of Habenaria dentata was also analyzed, so combined with your suggestions, the title of this manuscript was modified from ' Study on the Genetic Diversity of Germplasm Resources of Characteristic medicinal orchid plant Habenaria dentata ' to ' Study on the Genetic Diversity and Genetic Structure of Germplasm Resources of Characteristic medicinal orchid plant Habenaria dentata '. The second, in this manuscript, we preliminarily explored the reasons for the decrease of the population of Habenaria dentata, the possible endangered mechanism, and quickly determined the high-quality germplasm resources, which laid the foundation for other molecular marker techniques such as SSR. Therefore, only ISSR markers was used to analyze the genetic diversity of Habenaria dentata population in Guangxi, China. The third, in combination with your suggestions, this manuscript will supplement the relevant research in the discussion.

Reviewer 2 Report

The article "Study on the Genetic Diversity of Germplasm Resources of Characteristic medicinal orchid plant Habenaria dentata" used 8 ISSR primers out of 100 to study the genetic diversity between 133 samples representing six wild populations of Habenaria dentata. The work is bioneer. However, the number of 8 primers, is very low to construct the overall conclusion and relationships. I strongly recommend testing more primers to include more polymorphic primers.    

A few sentences are not correct because the structure was not true. 

Author Response

Dear reviewer,

  First of all thank you for your valuable comments. The purpose of this study is to preliminarily explore the reasons for the decrease in the population of Habenaria dentata and the possible endangered mechanism, and to quickly select high-quality germplasm resources. Therefore, only 100 universal primers were screened. In the subsequent experiments, we will further optimize the PCR reaction system and test more primers to draw a more comprehensive conclusion.

Reviewer 3 Report

The work presented for review " Study on the Genetic Diversity of Germplasm Resources of Characteristic medicinal orchid plant Habenaria dentata " concerns the genetic variation and differentiation of important terrestrial herb whose number of wild populations is decreasing dramatically. This type of research is important as it constitutes a starting point for the protection of genetic diversity, and the determination of particularly valuable populations that deserve protection.

I have just a few remarks listed below, that can improve the quality of the manuscript:

L 121-123: duplicated sentence

Paragraphs 3.1 and 3.2  contain methodological information, hence they should be found in the chapter Material and methods

L 180-184: It's one very long and complicated sentence. Should be reformulated.

In the Discussion there is no reference to genetic studies concerning the species Habenaria denta or even the genus Habenaria, and such are available in the scientific literature. In my opinion, the Discussion should be complemented by the results of these works.

Author Response

Dear reviewer,

  First of all, thank you for your valuable comments. In order to improve the quality of the manuscript, your proposed amendments have been made in this article. In the discussion, we have increased the study of related species in the discussion.

Round 2

Reviewer 1 Report

Dear authors,

Here are my comments:

Line 63. Authors should cite two other manuscripts on ISSR use, but, they should be from current and quality journals.

Lines 101-102, 120. Authors should indicate which stain they used for the agarose gel.

Lines 128-133. The authors in the methodology do not explain how they performed the structure analysis.

Lines 123-133. The authors do not indicate how they obtained the polymorphic information (PIC) and how they obtained the genetic distances (Mantel Test). The authors should indicate the PIC or PPL formula they used.

Lines 158 and 159. Regarding these results, the authors do not indicate in the item Materials and Methods how they obtained the percentage of polymorphic loci (PPL) and Shannon Information Diversity Index (I).

Lines 261-270. Citations 30,31,32,32,33 (and their corresponding references) are not indexed in WoS or Scopus. Manuscripts published in Journal of Zhejiang A & F University, Acta Botanica Boreali-Occidentalia Sinica, Acta Horticulturae Sinica and Oikos (among others) are of low quality.

Lines 274-280. Citations 34 and 35 and their references are more than 20 years old.

Lines 288-365. Citations 38,39, 40, 41, 41, 42, 43, 43, 44, 45, 46,47,48, 49, 50, 51, 52, 53, 54 and 55 do not appear in the References item. Sorry, but it is difficult to revise the manuscript this way.

Author Response

Dear Reviewer,

  First of all, thank you for your valuable advice. Now make the following reply to your question.

  Firstly, the Introduction of the article has been modified and the references cited have been updated. Secondly, this paper has supplemented the staining agent of agarose gel used in the material part. Thirdly, this paper has supplemented the references of genetic structure, genetic distance, polymorphism information and other related genetic information in the method. Fourth, the low-quality references have been updated.

  Finally, thank you again for your valuable comments.

Best wishes.

Reviewer 2 Report

Thanks for adding the new paragraphs in the discussion part. 

Author Response

Dear Reviewer,

  First of all, thank you for your valuable advice. Now make the following reply to your question.

  Compared with the previous manuscript, this manuscript has transferred some of the content under discussion to the Introduction.

  Finally, thank you again for your valuable comments.

Best wishes.

Reviewer 3 Report

L240/241 - double Fig. 5

L 270: „solve the relationship between genomes and genomes”  - there is a stylistic error here

In the Discussion section, the authors devoted much attention to the properties and use of ISSR markers. In my opinion, it is not necessary in such a deep approach, because these markers are already known and used in genetic research for many years.

Author Response

Dear Reviewer,

  First of all, thank you for your valuable advice. Now make the following reply to your question.

  Firstly, the problem of “double Fig.5”has been modified in the article. Secondly, ' solve the relationship between genomes and genomes ' has been deleted and modified. Thirdly, the discussion on ISSRs in the article has been streamlined and the discussion about ISSRs has been moved to the Introduction.

  Finally, thank you again for your valuable comments.

Best wishes.
